# Effectiveness of an mHealth Application to Overcome Problematic Smartphone Use: Comparing Mental Health of a Smartphone Control-Use Group and a Problematic-Use Group

**Mun Joo Choi** [1,2], **Sun Jung Lee** [1,2], **HyungMin Kim** [1,2], **Dai-Jin Kim** [3,*] **and In Young Choi** [1,2,*]

1  Department of Medical Informatics, College of Medicine, The Catholic University of Korea, Seoul 06591, Korea; cmj7705@naver.com (M.J.C.); lsjadd500@gmail.com (S.J.L.); hikhm0304@naver.com (H.K.)
2  Department of Biomedicine & Health Sciences, College of Medicine, The Catholic University of Korea, Seoul 06591, Korea
3  Department of Psychiatry, Seoul St. Mary's Hospital, College of Medicine, The Catholic University of Korea, Seoul 06591, Korea
*  Correspondence: kdj922@catholic.ac.kr (D.-J.K.); iychoi@catholic.ac.kr (I.Y.C.); Tel.: +82-2-2258-7586 (D.-J.K.); +82-2-2258-7870 (I.Y.C.)

**Abstract:** We developed an mHealth application that can help alleviate the problematic use of smartphones and allied psychological symptoms. This study observed the change in patterns of users' problematic smartphone use, depression, and anxiety while using the mHealth application. We conducted this study from 9 January to 10 April 2019. The Korean Smartphone Addiction Proneness Scale for Adults, Generalized Anxiety Disorder Scale, and the Patient Health Questionnaire were measured at week 0, 8, 12. A post hoc test of Repeated Measurement Anova analysis and Linear Mixed Model analysis were used. Overall, 190 participants were allocated into two groups. Sixty-six were in the smartphone control-use group and 124 were in the problematic-use group. The study elucidated the difference between the two groups in terms of problematic smartphone use and depression and anxiety after 13 weeks of using the mHealth application. This study showed the use of the mHealth application reducing problematic smartphone use scores and negative symptoms such as anxiety and depression in the PSU group. The results can be used as the basis for similar qualitative studies to further resolve the problematic use of smartphones.

**Keywords:** problematic smartphone use; depression; anxiety; mHealth; linear mixed model; digital therapeutics

## 1. Introduction

A recent survey on the ever-increasing use of smartphones worldwide over the last few years revealed that the global smartphone usage rate in 2020 was 45%, up from 38% in 2018. In countries such as the United States, the United Kingdom, Germany, and Sweden, the average smartphone ownership rate exceeded 75% [1]. According to a 2018 Pew Research Center survey, Korea's smartphone ownership rate is over 95%, which is the highest in the world [2].

As people's demand for smartphones increases, the usability and functions of smartphones have also increased over time, enabling social communication, increasing productivity in daily life, and assisting people in obtaining their informational, entertainment, and educational needs [3]. However, smartphone use increase and excessive usage are accompanied more likely with the negative effects of smartphones [4–7]. Particularly, the excessive usage of smartphones has decreased the self-ability to control the usage and causes the smartphone to be continuously used, which is causing maladjustment to daily life [8]. Furthermore, problematic smartphone use affects mental health, such as depression, anxiety [5,6,9,10], stress [5,10,11], and other issues.

Excessive use of smartphones can result in compulsive behavior, which can lead to problematic smartphone use [10,12–14]. Although excessive smartphone use is not yet classified as an addiction [15], it is defined as "problematic" smartphone use, with people presenting with symptoms similar to those of individuals with substance use disorder [11,16]. Problematic smartphone use is also related to mental health [17]. Prior research shows that it can affect mental health, leading to disorders such as depression and anxiety [9,11–14,17–19]. Therefore, there is a pressing need to manage and mitigate these problems.

Continuous intervention and self-management are both important for alleviating problematic smartphone use. Among the studies on the effectiveness of interventions on smartphone use, preliminary evidence in one study showed that these interventions can reduce symptoms of anxiety [20]. Mobile health (mHealth) interventions can assist in delivering effective health care by enabling access to personalized support [21,22]. Further, according to a recent study, it could positively affect motivation and achievement by using mobile-based devices [23]. Lin et al. (2017) proposed that an application-incorporated diagnostic tool has substantial accuracy for detecting problematic smartphone use [24]. The results showed that the smartphone application can support a self-control program and help implement an intervention and diagnosis for problematic smartphone use.

Therefore, we developed an mHealth application, called MindsCare, that performs an integrated intervention that can help alleviate the problematic use of smartphones and the allied psychological symptoms to assist in the treatment of the condition. The application aims to relieve problematic smartphone use. The user received personalized health care services, and it is designed to monitor smartphone usage, prevent overuse, diagnose the condition, and help in management of the condition. Problematic smartphone use and mental health symptoms are designed to be measured based on the designed intervention period through various psychological scales, and management or self-evaluation and intervention services are introduced.

The objective of this study was to observe the change in patterns of users' problematic smartphone use, their level of depression and anxiety while using MindsCare during the study period, and to observe the effect of MindsCare on smartphone usage and associated mental health. We aimed to achieve several specific goals. First, we wanted to determine the difference between each groups' problematic smartphone use, anxiety, and depression with reference to time, and we achieved this by dividing the users into a smartphone usage control group and a problematic smartphone use group. Second, changes in problematic smartphone use, depression, and anxiety between the two groups were compared to determine the group on which the application had the greatest influence. Therefore, we established primary and secondary study hypothesis as follows. Hypothesis 1: There are significant differences and changes in problematic smartphone use, anxiety, and depression scores between two groups. Hypothesis 2: There is a significant difference in time between the two groups, and the affecting factor will be gender.

## 2. Method

### 2.1. Study Design

The study was designed to analyze the effectiveness and changes in problematic smartphone use and psychological symptoms of users by adopting a smartphone usage management system (MindsCare). To identify the effects and changes, the study was designed and conducted as follows. The total experiment period was 13 weeks. Participants installed the MindsCare application at the 0th week of baseline and were divided into the Smartphone Usage Control (SUC) and Problematic Smartphone Use (PSU) groups, according to the results of the Korean Smartphone Addiction Proneness Scale for Adults (S-scale). Although participants were divided into two groups after conducting app survey 1, these two groups performed the same tasks during the intervention period. Then, the smartphone usage data up to the first week were collected. During this period, the participants autonomously used the MindsCare application and self-regulated the usage

of a smartphone without intervention. The intervention period was from weeks 1 to 8. During this period, participants' smartphone usage was controlled by us virtually by blocking the use through manual and automatic interventions (usage control alarms and encouraging messages, checking the survey results for self-evaluation, etc.). The ninth until the 12th weeks was the recurrence prevention period, which involved only maintaining the installed applications without any manual or automatic intervention from the researcher [25]. Assessments (problematic smartphone use, anxiety, depression) were conducted at baseline (0 week), the end of the intervention period (8 weeks), and at the end of the follow-up (12 weeks) period (See Figure 1).

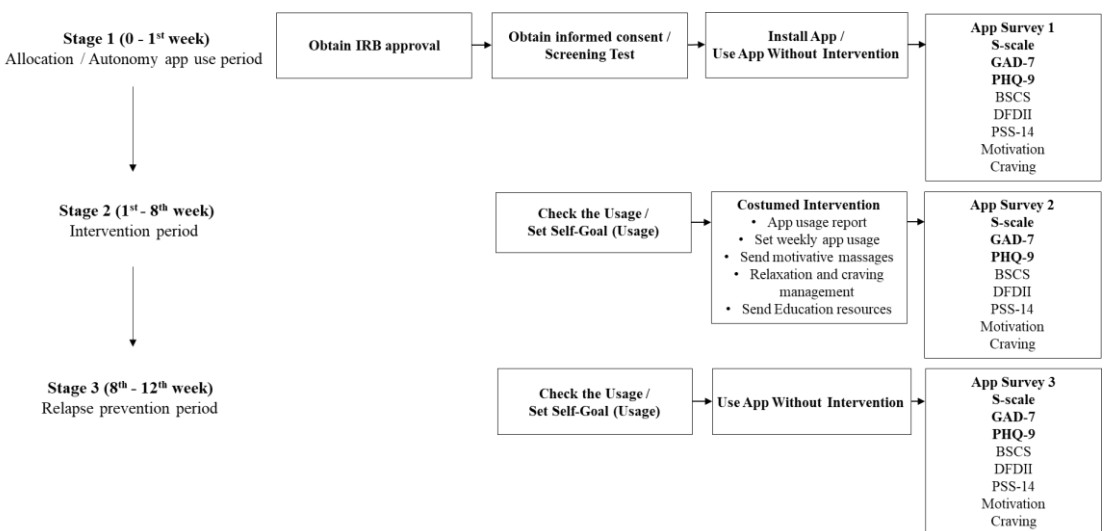

**Figure 1.** Study design.

Smartphone Addiction Proneness Scale (S-scale), Generalized Anxiety Disorder Scale (GAD-7), The Patient Health Questionnaire (PHQ-9), Brief Self-Control Scale (BSCS), Dickman Functional and Dysfunctional Impulsivity Inventory (DFDII), Perceived Stress Scale (PSS-14), Institutional Review Board (IRB).

*2.2. Participant Eligibility Criteria*

We conducted this study from 9 January through 10 April 2019, in South Korea. Participant recruitment was in collaboration with a professional internet survey agency in Korea. Five hundred users of smartphones over the age of 19 were randomly selected as the study population. All participants gave obtained, Informed consent online before participation. Those who did not meet the following inclusion criteria were excluded: (1) those who refused to provide consent, (2) users of Android Lollipop 5.0.1, and (3) those who used smartphones for less than 3 h a day. Further, during the experiment, those who did not take the survey at weeks 0, 8, and 12, those who did not maintain at least 85% of data collection time per day (24 h), and those who did not switch on the smartphone for more than 3 days per week were excluded. All participants were given incentives as follows: (1) those who filled out the app survey at week 0 received 1000 won (about $1); (2) those who filled out the app survey at week 8 received 1000 won (about $1); (3) those who filled out the app survey at week 12 received 4000 won (about $4); (4) those who maintained the app installation for 13 weeks and filled out all the app surveys at weeks 0, 8, and 12 received 10,000 won (about $10). Excluded participants were informed via online message or email and given incentives depending on the duration of their participation. Underage participants were excluded from the study, as parental consent was legally required. Additionally, the approval process of the Korean IRB is strict for underage participants. This study proceeded in accordance with the Declaration of Helsinki and the Institutional Review Board (IRB) of the Catholic University of South Korea, St. Mary's Hospital (MC18FNSI0020).

### 2.3. Participant Enrollment

On 9 January 2019, 500 individuals randomly participated in the study through a professional internet survey agency in Korea. Before conducting the survey for eligibility, potential participants were asked to fill in the consent form; 21 were excluded who did not agree to fill in the consent form. Among the rest of potential participants, 137 were excluded as they did not complete the baseline measurements ($n = 25$), were unable to access the application ($n = 11$), were not android users ($n = 27$); or used smartphones for less than 3 h a day ($n = 74$). Of the remaining 342, the S-scale was conducted, and the participants were divided into two groups. Between 0 week and 8 weeks, we excluded an additional 108 participants who did not fill out survey, were not connected to the application, or showed frequent missing data. From week 8 to 12, 24 participants were excluded, mostly because they did not fill out the survey. Further, another 20 were excluded after the follow-up period because of noncompliance with the usage measurement (i.e., collected less than 75% of usage measurement) or did not fill out the survey. Participants who did not complete the assessments were sent notifications to fill out the survey repeatedly. In week 0, 8, and 12, from Monday through Sunday, a maximum of 14 notifications were sent to the participants via SMS until they fill out the survey; when they did not, they were excluded. System errors that were not able to measure the usage and other errors were resolved via phone, email, and the inquiries function in the application with the administrator; the unresolved cases in which individual's device problems were identified were excluded from the study. Finally, in all, 190 participants were analyzed (see Figure 2 for a flow diagram).

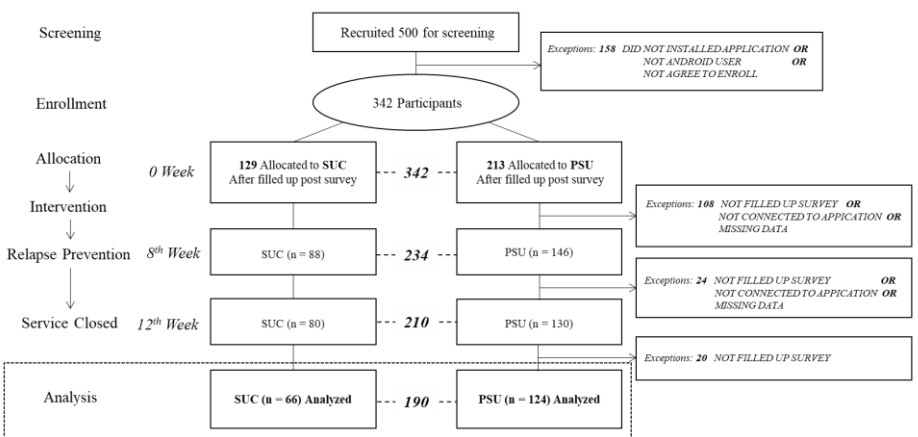

**Figure 2.** Participant enrollment flow diagram.

### 2.4. Study Tools and Measurements

#### 2.4.1. Smartphone Application MindsCare

In this study, we used a smartphone usage management application as a study tool. In a previous research we developed the Smartphone Overdependence Management System (SOMS), which is a background data collecting system, designed to deliver problematic use prevention and diagnosis based on scientific evidence [26]. The results of our previous study using SOMS showed that by using such an application data could identify problematic smartphone usage [27,28]. Therefore, SOMS was extended as a user-friendly interface application in MindsCare so that it can interact with users and provide intervention service. MindsCare is an android application that was developed for prevention of problematic smartphone use. The user received personalized health care services through the application. When the user installed the system on the smartphone, the application started to measure users' data relating to smartphone usage, estimated usage of downloaded applications, and saved usage records in real time. The users were able to set a daily usage goal for improving self-control, received educational image contents and relaxation therapy related to problematic smartphone use, were informed on local counseling centers

(I WILL center; internet addiction prevention counseling center in Korea), were provided messaging function to communicate with the treatment expert, and conduct a survey (on various psychological scales such as anxiety, depression, stress, impulsivity, self-control, problematic smartphone use, etc.) for self-evaluation at weeks 0, 8, and 12 through the MindsCare application. These functions are classified as automatic intervention. A web monitoring system was also developed to receive MindsCare users' data every 10 min. The usage pattern was analyzed from the stored usage time data, and the user could receive a personalized result analysis' service. The treatment expert provided personalized intervention by monitoring users' data from the developed website. These functions are classified as manual interventions. Therefore, MindsCare performed an integrated intervention function that can help alleviate the problematic use of smartphones and their allied psychological symptoms (See Figure 3).

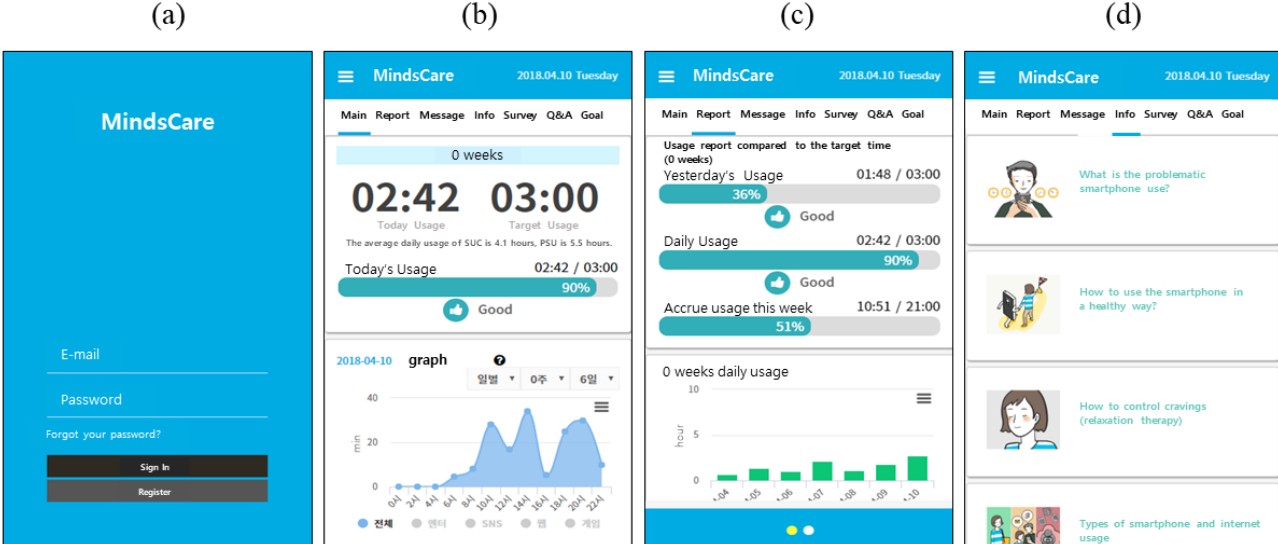

**Figure 3.** Screenshots of 'MindsCare' application. The features of MindsCare: (**a**) login page, (**b**) application's main screen, representing usage time of the day, goal, and daily and weekly usage graphs, (**c**) a more detailed report about usage time compared to self-targeted goal, and daily and weekly usage, (**d**) information about problematic smartphone use and ways to reduce craving.

### 2.4.2. Problematic Smartphone Use

We used the Korean Smartphone Addiction Proneness Scale for Adults (S-scale) [29] to measure problematic smartphone use based on the scores. The study tools comprised four sub-areas of resistance, including virtual world orientation, daily life disturbance, withdrawal, and tolerance. There were a total of 15 questions scored on a 4-point Likert scale with scores ranging from "Not at all (1)" to "Always (4)" The total scores were categorized into three groups following diagnostic criteria: high-risk groups, with a total score of 44 or above; potential-risk groups, with a total score of 40 to 43; and normal groups, with a score of 39 or below. Our study regrouped high-risk and at-risk group as PSU group (total score of 40 or above, mean: 45.99, standard deviation: 4.16) and the normal group as the SUC group (total score of 39 or below, mean: 34.59, standard deviation: 4.16). In this study, the measurement instruments (S-scale) had acceptable internal consistency (Cronbach's alpha = 0.797).

### 2.4.3. Anxiety

Generalized Anxiety Disorder (GAD) scale is a practical self-report anxiety questionnaire that has been proven valid in primary care to assess the severity of GAD symptoms [30]. GAD is characterized for those excessively anxious, not limited to certain circumstances. The GAD-7 includes seven questions scored on a 3-point Likert scale from

"Not at all (0)" to "Nearly every day (3)". Higher scores indicate higher severity. The total scores are categorized into four subgroups: Severe 15 to 21; Moderate 10 to 14; Mild 5 to 9; None 0 to 4. In this study, the measurement instruments (Korean version of the GAD-7) had excellent internal consistency (Cronbach's alpha = 0.921).

### 2.4.4. Depression

The Patient Health Questionnaire (PHQ) is a scale to assess depression. PHQ-9 comprises nine criteria on which the diagnosis of DSM-5 depressive disorders is based. There were a total of nine questions scored on a 3-point Likert scale ranging from "Not at all (0)" to "Nearly every day (3)." Scores are categorized into five subgroups: Severe 20 to 27; Moderate severe 15 to 19; Moderate 10 to 14; Mild 5 to 9; None 0 to 4 [31]. In this study, the measurement instruments (Korean version of the PHQ-9) had excellent internal consistency (Cronbach's alpha = 0.908).

### 2.5. Statistic Analysis

We conducted an analysis to compare scores at three time points of baseline 0 week, after intervention period of 8 weeks, and after follow-up period at 12 weeks on S-scale, GAD, and PHQ scores between SUC and PSU. To compare whether there was a significant difference in changes by time point in the two groups, a post hoc test of Repeated Measurement Anova (RM Anova) analysis was conducted using Bonferroni's method. Through Linear Mixed Model (LMM) analysis, we compared the difference in the pattern of change over time between the two groups. LMM is a representative statistical method used in longitudinal data analysis and is typically used to understand changes in human behavior over time [32,33]. Therefore, we used LMM method because of the study design, and the aim of the study was to observe the significant difference and change in problematic smartphone use, anxiety, and depression between the two groups according to the period of the MindsCare use. To identify the socio-demographic characteristics, the frequency, and percentage of the two groups, a chi-square test was conducted to verify homogeneity of variance. Statistical analyses, descriptive analyses, and *t*-tests for the two groups were conducted with IBM SPSS Statistics for windows, version 24.0.

## 3. Results

### 3.1. Demographics

The demographic results were as shown in Table 1. Among the 190 participants, 65% were in the PSU group and 35% were in the SUC group. Overall, 47% were men, 52% women, 33% were 19 to 29 years old, 37% were 30 to 39 years, and 30% were over 40. The most frequently used applications in the past year (self-reported) were 20 web surfing in the SUC (20) and Social Network Service (SNS) and web surfing in PSU group (37). The *p* value of all variables in both groups was greater than 0.05. The *p* value of gender between the groups was 0.822, age group was 0.593, marital status was 0.171, education was 0.901, occupation was 0.884, and most used application was 0.944. Hence, there was no statistically significant difference.

**Table 1.** Demographic data of SUC and PSU.

| Characteristics | | SUC (*n* = 66) | PSU (*n* = 124) | X$^2$ Value | *p*-Value |
|---|---|---|---|---|---|
| Gender | Male | 32 (35.6) | 58 (64.4) | 0.051 | 0.822 |
| | Female | 34 (34.0) | 66 (66.6) | | |
| Age group | 19 to 29 | 24 (38.7) | 38 (61.3) | 1.045 | 0.593 |
| | 30 to 39 | 25 (35.2) | 46 (64.8) | | |
| | Over 40 | 17 (29.8) | 40 (70.2) | | |

**Table 1.** *Cont.*

| Characteristics | | SUC (*n* = 66) | PSU (*n* = 124) | X² Value | *p*-Value |
|---|---|---|---|---|---|
| Marital status | Married | 44 (38.6) | 70 (61.4) | 1.873 | 0.171 |
| | Unmarried | 22 (28.9) | 54 (71.1) | | |
| Education | High School or Lower | 7 (35.0) | 13 (65.0) | 0.208 | 0.901 |
| | College Student | 10 (31.3) | 22 (68.8) | | |
| | Graduate School or Above | 49 (35.5) | 89 (64.5) | | |
| Occupation | White-collar/Professional | 29 (33.7) | 57 (66.3) | 0.653 | 0.884 |
| | Student | 10 (31.3) | 22 (68.8) | | |
| | Unemployed | 11 (40.7) | 16 (59.3) | | |
| | Others | 16 (35.6) | 29 (64.4) | | |
| Most Used Application for the Past One Year | SNS | 17 (31.5) | 37 (68.5) | 0.761 | 0.944 |
| | Web Surfing | 20 (35.1) | 37 (64.9) | | |
| | Game | 9 (40.9) | 13 (59.1) | | |
| | Entertainment | 8 (38.1) | 13 (61.9) | | |
| | Others | 12 (33.3) | 24 (66.7) | | |

Values are presented as number (%). The *p*-values were obtained by chi-square test. Statistically significant associations (*p* < 0.05), Pearson Chi-Square (X² value), Smartphone Use Control (SUC), Problematic Smartphone Use (PUC), and Social Network Service (SNS).

*3.2. Outcomes*

First, we assessed the fixed effect of the two groups (SUC = 1, PSU = 2) from the results of the 0 week S-scale survey total scores. Then, to compare significant differences and changes by time point (0 week, 8 weeks, 12 weeks) of the two groups, a post hoc test of RM Anova was conducted using Bonferroni's method. The post hoc test of RM Anova tests the mean difference between variables, highlighting exactly the point where these differences lie, which is not possible through RM Anova analysis [34]. At this time, we used *p*-value corrected by Bonferroni's method (Corrected significance level = 0.05/3 = 0.016). Through this analysis, we examined the differences between the two groups in terms of problematic smartphone use, depression, and anxiety depending on the period of using MindsCare and its trend (change over time) for 0 to 12 weeks (13 weeks in total).

Longitudinal data analysis is a method of analyzing data observed at regular intervals for the same object, and the analysis of such data is called LMM [32]. Therefore, through LMM analysis, we examined the difference between the two groups, with changes in S-scale (problematic smartphone use), PHQ (depression), and GAD (anxiety) for 0, 8, and 12 weeks in each group while using MindsCare.

3.2.1. Problematic Smartphone Use

After measuring the average score of the S-scale total score for 0 weeks, 8 weeks, and 12 weeks, we conducted the post hoc test of the RM Anova to compare significant differences between the two groups, using *p*-value with Bonferroni correction. In the SUC group, no significant differences were seen in point-in-time comparisons (0 week to 8 weeks, 8 weeks to 12 weeks, all *p*-value = 1.000), and in the PUS there was a significant decrease over time (0 week to 8 weeks, *p*-value = 0.000/8 week to 12 weeks, *p*-value = 0.018). In addition, LMM analysis was conducted for comparison of intergroup effect sizes using regression coefficient beta along with differences in factors, such as gender. As a result, the difference in the total S-scale scores by gender had a *p*-value of 0.577 and, hence, no significant differences were seen (See Table 2). However, significant differences in S-scale

were seen for group, time, and group × time. The LMM analysis showed that the patterns of change over time between the two groups differed. The estimate at [Group = 1] × time was 2.865, which was shown to increase the sum of S-scale scores by 2.865 over one time in the SUC when compared to the PUS. In other words, the sum of the S-scale of PUS was shown to decrease more over time when compared to SUC (See Table 3).

**Table 2.** Results of post hoc analysis (S-scale, GAD, PHQ).

| | | Post-Hoc Test | | Mean Diff. (I–J) | Std. Error | Sig. * | CI | |
|---|---|---|---|---|---|---|---|---|
| | **Group** | **(I) Time** | **(J) Time** | | | | **Lower Bound** | **Upper Bound** |
| S-scale | SUC | 0 week | 8 weeks | 0.803 | 0.678 | 0.712 | −0.829 | 2.435 |
| | | 0 week | 12 weeks | 0.561 | 0.862 | 1.000 | −1.519 | 2.641 |
| | | 8 weeks | 12 weeks | −0.242 | 0.678 | 1.000 | −1.874 | 1.389 |
| | PSU | 0 week | 8 weeks | 4.919 * | 0.495 | <0.001 | 3.729 | 6.110 |
| | | 0 week | 12 weeks | 6.290 * | 0.629 | <0.001 | 4.773 | 7.808 |
| | | 8 weeks | 12 weeks | 1.371 * | 0.495 | 0.018 | 0.181 | 2.561 |
| GAD | SUC | 0 week | 8 weeks | 0.303 | 0.504 | 1.000 | −0.909 | 1.515 |
| | | 0 week | 12 weeks | 0.076 | 0.552 | 1.000 | −1.258 | 1.409 |
| | | 8 weeks | 12 weeks | −0.227 | 0.504 | 1.000 | −1.439 | 0.985 |
| | PSU | 0 week | 8 weeks | 1.815 * | 0.367 | <0.001 | 0.930 | 2.699 |
| | | 0 week | 12 weeks | 2.258 * | 0.403 | <0.001 | 1.285 | 3.231 |
| | | 8 weeks | 12 weeks | 0.444 | 0.367 | 0.685 | −0.441 | 1.328 |
| PHQ | SUC | 0 week | 8 weeks | 1.091 | 0.620 | 0.238 | −0.401 | 2.583 |
| | | 0 week | 12 weeks | 0.424 | 0.637 | 1.000 | −1.113 | 1.962 |
| | | 8 weeks | 12 weeks | −0.667 | 0.620 | 0.849 | −2.159 | 0.825 |
| | PSU | 0 week | 8 weeks | 2.210 * | 0.452 | <0.001 | 1.121 | 3.298 |
| | | 0 week | 12 weeks | 2.927 * | 0.465 | <0.001 | 1.806 | 4.049 |
| | | 8 weeks | 12 weeks | 0.718 | 0.452 | 0.340 | −0.371 | 1.806 |

Statistically significant associations (* $p < 0.05$), Bonferroni correction were used, Confidence interval (CI), Smartphone Use Control (SUC), Problematic Smartphone Use (PUC), Smartphone Addiction Proneness Scale (S-scale), Generalized Anxiety Disorder Scale (GAD), The Patient Health Questionnaire (PHQ).

**Table 3.** Results of linear mixed model (gender, group, time, group × time).

| Dependent Variable | Fixed Effects | numDF | denDF | *F*-Value | *p*-Value |
|---|---|---|---|---|---|
| S-scale | (Intercept) | 1 | 374.982 | 3914.799 | <0.001 |
| | Gender | 1 | 186.991 | 0.313 | 0.577 |
| | Group | 1 | 374.703 | 110.413 | <0.001 |
| | Time | 1 | 207.583 | 40.763 | <0.001 |
| | Group × Time | 1 | 207.583 | 28.512 | <0.001 |
| GAD | (Intercept) | 1 | 385.172 | 270.135 | <0.001 |
| | Gender | 1 | 186.912 | 0.294 | 0.588 |
| | Group | 1 | 385.148 | 27.801 | <0.001 |
| | Time | 1 | 204.051 | 11.632 | 0.001 |
| | Group × Time | 1 | 204.051 | 10.171 | 0.002 |

**Table 3.** *Cont.*

| Dependent Variable | Fixed Effects | numDF | denDF | *F*-Value | *p*-Value |
|---|---|---|---|---|---|
| | (Intercept) | 1 | 383.295 | 380.835 | <0.001 |
| | Gender | 1 | 186.879 | 0.023 | 0.879 |
| PHQ | Group | 1 | 383.243 | 27.707 | <0.001 |
| | Time | 1 | 203.621 | 18.050 | <0.001 |
| | Group × Time | 1 | 203.621 | 10.068 | 0.002 |

Statistically significant associations ($p < 0.05$), Degree of Freedom (DF), Smartphone Addiction Proneness Scale (S-scale), Generalized Anxiety Disorder Scale (GAD), The Patient Health Questionnaire (PHQ).

### 3.2.2. Anxiety

Based on the measurement of the average score of the GAD total score for 0 week, 8 weeks, and 12 weeks, the SUC group showed no significant difference between points in the post hoc analysis, while the PUS group showed a significant decrease: From 0 week to 8 weeks, *p*-value was 0.000, and between 8 weeks to 12 weeks, the *p*-value was 0.685, which indicated that GAD total scores were maintained without decreasing (See Table 2). The LMM analysis results showed no difference in GAD total score by gender, but significant difference in GAD total score by group, time, and group × time. The estimate at [Group = 1] × time was 1.0912, indicating that the GAD total score increased by 1.091 over one time when compared to the PUS. In other words, compared to the SUC, the sum of GAD scores in the PUS was shown to decrease more over time (See Table 3).

### 3.2.3. Depression

Based on the measurement of the average score of the PHQ total score for 0 week, 8 weeks, and 12 weeks, we conducted a post hoc test and LMM analysis, demonstrating that the SUC group showed no significant difference between time points, and the PUS group showed a significant decrease between 0 week to 8 weeks, *p*-value 0.000, and between 8 weeks to 12 weeks, when *p*-value was 0.340, resulting in the PHQ scores remaining unchanged (See Table 2). LMM analysis showed that there was no difference in the total score of PHQ by gender, similar to the anxiety results, and there was a significant difference in the total PHQ score in the three categories: group, time, and group × time. The estimate at [Group = 1] × time was 1.2516, indicating that the SUC increased by 1.252 over time compared to the PUS. In other words, the sum of PHQs of PUS was shown to decrease more over time than that of SUC (See Table 3).

## 4. Discussion

### 4.1. Principal Findings

The study objective was to investigate the change in patterns of users' problematic smartphone use and their level of anxiety and depression while using MindsCare during the study period. This was done to confirm the effect of MindsCare on problematic smartphone usage associated with mental health and to identify the effectiveness of the application on the problematic user group.

The results revealed the difference between the two groups in terms of problematic smartphone use and depression and anxiety after a total of 13 weeks of MindsCare use. First, there was no significant change in the SUC group for problematic smartphone use, while in the PSU group, problematic smartphone use scores were reduced to the potential-risk group score from 0 to 8 weeks and it showed a continuous decrease until week 13, nearly reaching the normal group's scores. These results demonstrated that smartphone usage management apps such as MindsCare were effective on the PSU group. Likewise, there was no significant change in the SUC group for GAD, while in the PSU group, there was no change in the subgroup, but it showed a continuous decrease in scores until week 13. Interestingly, it showed a similar pattern in PHQ. There was no significant

change in the SUC group, but the PSU group showed a continuous decrease in scores until week 13. Similar to prior research results, this can be attributed to our mainly designing the interventions focusing on reducing on problematic smartphone use (See Figure 4) [35].

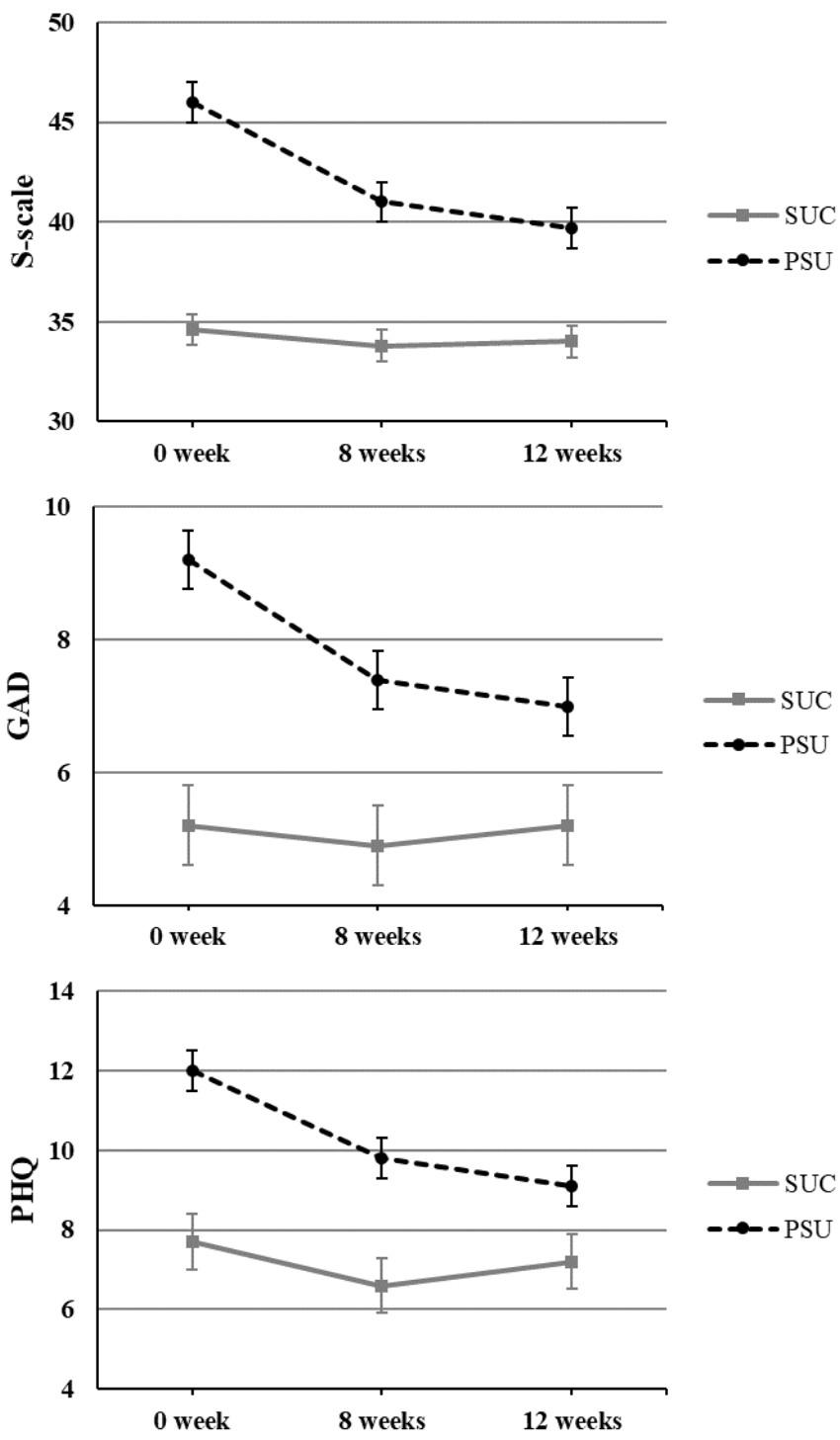

**Figure 4.** Results of linear mixed effects models of the post hoc analysis (S-scale, GAD, PHQ).

### 4.2. Strengths, Implications, and Limitations

Lin et al. (2017) suggested that application-integrated diagnostic tools may have significant accuracy in detecting problematic smartphone use and identifying problematic smartphone use based on data recorded in psychiatric patient interviews and apps [24]. However, our work is distinct in that we measured problematic smartphone use, anxiety,

and depression and how these symptoms can be reduced through manual and automatic intervention by continuously providing programs to alleviate depression or anxiety rather than simply reducing usage time.

Prior research has shown that using applications for mental health have been found to reduce anxiety and depression and interventions can be effective on problematic smartphone use [20,24] when associated with anxiety and depression [11]. The results of this study showed that integrated interventions using these application aspects are effective. Although no prior relationship was noted, it was predicted that a relationship existed connecting problematic smartphone use, depression, and anxiety, but this is a topic for future study.

The unique findings in our research support our research goal in that both continuous intervention and self-management are important to mitigate problematic smartphone use. While conducting the study, the proportion of problematic smartphone users was higher than that of normal people, although there were no artificial constraints to recruit problematic smartphone users when conducting the study enrollment.

The limitations of this study are that the findings cannot be generalized to the entire population and are limited to adults, due to the difficulty in gaining access to IRB and consent from non-adults. According to recent research by Kalogiannakis et al. (2020), the use of mobile devices increases utility and efficiency, which increases the learning effect compared to other devices for young children [36]. Since existing studies show that adolescents are relatively more vulnerable to smartphone use [37,38], future research on adolescents using a similar research protocol based on the results of this study would be valuable. Furthermore, despite numerous studies from various perspectives, "problematic smartphone use" has not yet been classified as an "addiction." Therefore, interventions that match the therapeutic effects of problematic smartphone use and the factors that cause them are difficult to establish.

## 5. Conclusions

This study showed changes in problematic smartphone use, depression, and anxiety patterns between the two groups during the intervention period while using a smartphone usage management application. Between the two groups, the MindsCare was more effective in the PSU group. Therefore, we proved the effectiveness of MindsCare in reducing problematic smartphone use scores and negative symptoms such as anxiety and depression in the PSU group. Since the problematic smartphone use also indicated that it may be related to mental health such as anxiety and depression, MindsCare may be useful as an approach that can address all of these areas comprehensively. These results can be used as the basis for similar qualitative studies to further resolve the problematic use of smartphones. Although the study was restricted to Koreans, its findings can be expanded and applied to other countries, considering national characteristics.

**Author Contributions:** Conceptualization, M.J.C.; Data curation, M.J.C. and S.J.L.; Formal analysis, M.J.C. and H.K.; Funding acquisition, D.-J.K.; Investigation, I.Y.C.; Methodology, M.J.C.; Project administration, D.-J.K. and I.Y.C.; Supervision, I.Y.C.; Validation, M.J.C. and H.K.; Visualization, M.J.C.; Writing—original draft, M.J.C.; Writing—review and editing, M.J.C. and I.Y.C. All authors had full access to all data and contributed to the manuscript. All authors have read and agreed to the published version of the manuscript.

**Funding:** This work was supported by the National Research Foundation of Korea (NRF) grant funded by the Korea government (MSIT) (No.2019R1A5A2027588).

**Institutional Review Board Statement:** This study proceeded in accordance with the Declaration of Helsinki and the Institutional Review Board (IRB) of the Catholic University of South Korea, St. Mary's Hospital (MC18FNSI0020).

**Informed Consent Statement:** The requirement for written, informed consent was waived by the Research Ethics Committee of the Catholic Medical Centre and this study was performed in accordance with relevant guidelines and regulations.

**Data Availability Statement:** Data sharing was not applicable to this study.

**Conflicts of Interest:** The authors declare no conflict of interest.

**Abbreviations**

mobile Health (mHealth); Smartphone Usage Control (SUC); Problematic Smartphone Use (PSU); Institutional Review Board (IRB); Smartphone Overdependence Management System (SOMS); Korean Smartphone Addiction Proneness Scale for Adults (S-scale); Generalized Anxiety Disorder (GAD); Patient Health Questionnaire (PHQ); Repeated Measurement Anova (RM Anova); Linear Mixed Model (LMM); Social Network Service (SNS).

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
