# Peer review of "Effectiveness of an mHealth Application to Overcome Problematic Smartphone Use: Comparing Mental Health of a Smartphone Control-Use Group and a Problematic-Use Group"

_applsci, doi:10.3390/app11188716_

Round 1
Reviewer 1 Report
The work concerns a very interesting and innovative aspect of the use of mobile applications in health care. The topic is very important, especially in the field of mental health. Everyone has a smartphone today and the use of modern technologies in health monitoring seems to be justified.
Weaknesses. A very poor discussion significantly lowers the level of publication. It would be justified for the authors to enrich this part of the work, which would internationalize the reception of the publication.
Author Response
Dear Reviewer,
I appreciate the time and effort that you have dedicated to providing your valuable feedback on my manuscript. I am grateful about your insightful comments on my paper. According to your advice, I have enriched the discussion part. Thank you for your insightful comments on my paper again (please see the attachment for the revised manuscript).

Reviewer 2 Report
Dear author(s)
My personal opinion is that after reading the paper, the manuscript is of potential interest to the readership of this journal, but there are significant issues that must be addressed before the article could be published:
In general:
- Background – Expand a little more to highlight the research problem to highlight the study's need.
- Methodology - expand a little more. Add analysis methods.
- Contribution: It would be a good paper if it did look at the research impact on the community.
- Findings: Should align with the study goal.
- Recommendations: Expand a little more.
- Recommendation for Researchers
- Introduction
Your introduction would be improved by the following:
- state the research contributions more clearly
- align the abstract to the contents of your article
- be precise about how the research was conducted
- identify the research gap and the goals of the research
- describe knowledge acquisition behaviour concisely
- correct grammar and punctuation errors
- Literature Review
Your literature review would be improved by the following:
- Use more recent studies
- Establish your main research question in a logical way
- Discuss the research questions and link these to your research gap and to the factors in the research model
3 Your discussion section would be improved by the following:
- In the discussion section is where you compare the results you have from the data analysis with the results from the literature you have reviewed, considering each hypothesis in turn
- You also need to explain what is new about your findings
- You also need to discuss what new knowledge has been added in this domain
- You also need to correct all grammar, spelling, and punctuation errors.
- 4. Conclusion
Your conclusion would be improved by the following:
- In your conclusion you need to include the newly formulated theoretical contributions
Abstract
In your abstract you need to summarise your research precisely, articulating research contributions clearly, making sure that you include a summary of all elements of the research including the variables tested by the hypotheses. Make sure you include how sampling was done. You should also elaborate on managerial implications for tertiary education during a pandemic.
Overall, the paper requires more information and focus. The areas requiring attention are highlighted in the individual sections.
In summary, the paper needs
- a re-write of the abstract to give a good summary of the paper and mention the key concepts.
- Expanding the introduction by clearly stating the research problem to suitably inform the reader.
- A synthesised and structured critique of the literature.
- clarifying the research procedures with an adequate explanation of the methods.
- Improving the survey instrument – construction of the questionnaire and the validity tests.
- Expanding the discussion to allow writing a well-developed conclusion summarising the entire paper. The outcomes should be discussed in relation to the existing research.
- emphasizing the significance of the research - a clear showing of how the findings contribute to new knowledge.
- Using results to support the claims in conclusion adequately, and how the results of the research can be used for future research
- A more recent bibliography is necessary. Remove all outdated references! Furthermore, the reference list of new publications is a little bit weak. There are not enough studies from Europe or western countries. Before I can make a final decision on the paper, please refer to more references. It is suggested that the author(s) can consider the following papers related to mobile device usage etc. to strengthen the background and conclusions of the study:
- Drolia, M.; Sifaki, E.; Papadakis, S.; Kalogiannakis, M. (2020). An Overview of Mobile Learning for Refugee Students: Juxtaposing Refugee Needs with Mobile Applications’ Characteristics. Challenges,11(2),31.
- Papadakis, S., & Kalogiannakis, M. (2020). A research synthesis of the real value of self-proclaimed mobile educational applications for young children. Mobile learning applications in early childhood education, 1-19.
In general, the English in the present manuscript is of publication quality and requires minor improvement. Please carefully proof-read spell check to eliminate grammatical errors
Plagiarism check results:
/* Similarity check with iThenticate revealed a similarity index of 21%, which is considered not appropriate. A maximum of around 60 quoted words is accepted per paper. There are papers with over 60 words. No previously copyrighted material was used. See the attached file for more details.
In preparing a revised manuscript, please also include a table of how you have responded to each of the issues listed above point by point.
Dear AUTHORS, summarizing my feedback, I expect your contribution to be highly valued by the journal's readers if you improve it according to the review statements. Please resubmit for review within a month at the latest.
I look forward to receiving your revised manuscript shortly.
With best regards

Author Response
Dear Reviewer,
We deeply appreciate the time and effort that you have dedicated to providing your valuable feedback on our manuscript. We are grateful about your insightful comments on the manuscript. According to your advice, we were able to incorporate changes to reflect most the suggestions and indicated the changes by using tracked changes within the manuscript. Here is a point-by-point response to your comments and concerns.
0. In general:
- Background – Expand a little more to highlight the research problem to highlight the study's need.
- Methodology - expand a little more. Add analysis methods.
- Contribution: It would be a good paper if it did look at the research impact on the community.
- Findings: Should align with the study goal.
- Recommendations: Expand a little more.
- Recommendation for Researchers
0. Response
Thank you for your accurate summarization. We will carefully revise the manuscript with your valuable advice.
1. Introduction
Your introduction would be improved by the following:
state the research contributions more clearly
align the abstract to the contents of your article
be precise about how the research was conducted
identify the research gap and the goals of the research
describe knowledge acquisition behaviour concisely
correct grammar and punctuation errors
1. Response
This is very important comment of you, because the introduction is the “first impression” of our article to the readers. Therefore, we correct the grammar and punctuation errors, and includes implication if study problems in liens 45~56. Also refer more references to strengthen the background of the study in lines 74~76.
2. Literature Review
Your literature review would be improved by the following:
Use more recent studies
Establish your main research question in a logical way
Discuss the research questions and link these to your research gap and to the factors in the research model
2. Response
Thank you for providing us such a productive comment. We refer more recent references to strengthen the literature review in lines 54, and 74~76. According to your advice, we set the study hypothesis in lines 282~287 to establish research question in a logical way.
3. Your discussion section would be improved by the following:
In the discussion section is where you compare the results you have from the data analysis with the results from the literature you have reviewed, considering each hypothesis in turn
You also need to explain what is new about your findings
You also need to discuss what new knowledge has been added in this domain
You also need to correct all grammar, spelling, and punctuation errors.
3. Response
We carefully revised the manuscript line by line again and tried to solve the inconsistency in the manuscript regarding the grammar, spelling, and punctuation errors. Thank you for mentioning this problem and giving us the opportunity to solve it. Also we changed the structure of the manuscript, moved from conclusion to discussion and included recent researches for comparison with the study results.
4. Conclusion
Your conclusion would be improved by the following:
In your conclusion you need to include the newly formulated theoretical contributions
4. Response
Again, thank you for your valuable comment. We re-white the conclusion according to your advice. Please refer in lines 537~550.
5. Abstract
In your abstract you need to summarise your research precisely, articulating research contributions clearly, making sure that you include a summary of all elements of the research including the variables tested by the hypotheses. Make sure you include how sampling was done. You should also elaborate on managerial implications for tertiary education during a pandemic.
Overall, the paper requires more information and focus. The areas requiring attention are highlighted in the individual sections.
5. Response
We truly agree with your advice. Our abstract was not well organized. Therefore, we summarized the research abstract precisely following our research goals. Please refer in lines 17~35.
6. In summary, the paper needs
a re-write of the abstract to give a good summary of the paper and mention the key concepts.
Expanding the introduction by clearly stating the research problem to suitably inform the reader.
A synthesised and structured critique of the literature.
clarifying the research procedures with an adequate explanation of the methods.
Improving the survey instrument – construction of the questionnaire and the validity tests.
Expanding the discussion to allow writing a well-developed conclusion summarising the entire paper. The outcomes should be discussed in relation to the existing research.
emphasizing the significance of the research - a clear showing of how the findings contribute to new knowledge.
Using results to support the claims in conclusion adequately, and how the results of the research can be used for future research
A more recent bibliography is necessary. Remove all outdated references! Furthermore, the reference list of new publications is a little bit weak. There are not enough studies from Europe or western countries. Before I can make a final decision on the paper, please refer to more references. It is suggested that the author(s) can consider the following papers related to mobile device usage etc. to strengthen the background and conclusions of the study:
Drolia, M.; Sifaki, E.; Papadakis, S.; Kalogiannakis, M. (2020). An Overview of Mobile Learning for Refugee Students: Juxtaposing Refugee Needs with Mobile Applications’ Characteristics. Challenges,11(2),31.
Papadakis, S., & Kalogiannakis, M. (2020). A research synthesis of the real value of self-proclaimed mobile educational applications for young children. Mobile learning applications in early childhood education, 1-19.
In general, the English in the present manuscript is of publication quality and requires minor improvement. Please carefully proof-read spell check to eliminate grammatical errors.
6. Response
We really appreciate your insightful comments. And also thank you so much for your effort to enrich our manuscript. Again, as your advice, we include a recent bibliography in the introduction as well as in the discussion. Also, we tried our best to check the typo and eliminate grammatical errors. Furthermore, we improved the survey instrument by construction, validity tests in lines 206~201, 271~272, 280~282.
7. Plagiarism check results:
/* Similarity check with iThenticate revealed a similarity index of 21%, which is considered not appropriate. A maximum of around 60 quoted words is accepted per paper. There are papers with over 60 words. No previously copyrighted material was used. See the attached file for more details.
In preparing a revised manuscript, please also include a table of how you have responded to each of the issues listed above point by point.
7. Response
This is very important point that thankfully provided. We checked similarity with iThenticate revealed a similarity index of 17% (we exclude the bibliography, please see the attachment) most of them overlapped from my former work, even though I try my best to avoid the similarity.

Reviewer 3 Report
The paper reports the effectiveness of a mobile application to overcome problematic smartphone use. Overall, the review of this work is favorable, and the paper focuses on an important topic. Having said that, there are a few shortcomings that currently prevent the paper from fulfilling its full potential. I hope the authors will find my suggestions helpful.
The authors wrote an interesting paper about the effectiveness of a smartphone application to overcome problematic smartphone use. However, it needs improvements and clarifications to be accepted. I recommend revision to remedy these major concerns, as well as some other less global issues.
Comment 1
The study frames partially the issues being studied. The problematic smartphone uses and implications of this problematic use need to be explained further.
Comment 2
A detailed description of how the application works and what are actions that the user can do with it is needed.
Comment 3
Although the title of the manuscript is quite clear that the study seeks for assessing the effectiveness of the smartphone application, the aims are not clearly defined. Effectiveness should be defined, and primary and secondary hypotheses need to be stated in a testable form.
Comment 4
Recruitment of users and allocation is not clear and need further clarification. Are the users allocated randomly? I do not think so, since the sample size of each group is not the same.
Comment 5
What is the difference between the Smartphone Usage Control group and the Problematic Smartphone Use group? Did the users need to perform different tasks during the intervention period?
Comment 6
Please, explain the incentives that participants received.
Comment 7
The reliability of the scales and questionnaires applied should be provided.
Comment 8
Statistical analyses need to be improved. A description of the analytic strategy for answering each research question should be provided. Covariates should be considered. To carry out a two-way mixed ANCOVA or moderation and mediation analyses are recommended.
Comment 9
Some information reported in the result's section should be reported in the method (e.g., lines 237-256; 280-288)
Comment 10
Line 249: “Participants who did not complete the assessment were sent notifications to fill up the survey repeatedly and when did not, they were excluded”. Please, provided quantitative data about how many notifications were sent before exclusion. How many participants were excluded for this or other reasons? An intention-to-treat analysis may be conducted.
Comment 11
There is no discussion, just a summary of the results. A discussion of the implications of the results in terms of substantive findings, and taking into account sources of potential bias and threats to internal and statistical validity, is needed. Moreover, similarities and differences between reported results and previous studies should be included.
Comment 12
Minor issues.
Table 1. Statistics need to be reported. Correct age group data.
Acronym SNS should be defined.
The notation 0.000 needs to be changed by < 0.001
Author Response
Dear Reviewer,
First of all, thank you for giving me the opportunity to a revised draft of my manuscript. I appreciate the time and effort that you have dedicated to providing your valuable feedback on my manuscript. I am grateful about your insightful comments on my paper. I have been able to incorporate changes to reflect most the suggestions and indicated the changes by using tracked changes within the manuscript. Here is a point-by-point response to your comments and concerns (please see the attachment for the revised manuscript).
Point 1.
The study frames partially the issues being studied. The problematic smartphone uses and implications of this problematic use need to be explained further.
Response 1.
Thank you for your valuable comments. We have improved introduction part, explained the problematic smartphone uses and implications of this problematic use on page 2, in lines 46~58.
Point 2
A detailed description of how the application works and what are actions that the user can do with it is needed.
Response 2.
Thank you for your suggestion. I thereby describe how the application works and what are the actions that the user can do in detail. Please refer at page 6, in Line 211~227.
Point 3
Although the title of the manuscript is quite clear that the study seeks for assessing the effectiveness of the smartphone application, the aims are not clearly defined. Effectiveness should be defined, and primary and secondary hypotheses need to be stated in a testable form.
Response 3.
A very important point that must be handled that you have thankfully provided. To propose clear aims of our research, we stated the primary and secondary hypotheses at page 7, in Line 277~283.
Point 4.
Recruitment of users and allocation is not clear and need further clarification. Are the users allocated randomly? I do not think so, since the sample size of each group is not the same.
Response 4.
You are absolutely right. There are mismatched in the figure 2 and manuscript about the recruitment of users and allocation. Five-hundred individuals were randomly enrolled in the study, then 158 participants were excluded. 342 participants were allocated into two groups after conducting app survey 1(according to the result of s-scale). We corrected the manuscript and the figure 2 accordingly.
Point 5.
What is the difference between the Smartphone Usage Control group and the Problematic Smartphone Use group? Did the users need to perform different tasks during the intervention period?
Response 5.
Another very sharp comment that you have thankfully provided. We divided Smartphone Usage Control group and the Problematic Smartphone Use group, according to the scores of the Korean Smartphone Addiction Proneness Scale for Adults (S-scale). The total scores were categorized into three groups following diagnostic criteria: High-risk groups, with a total score of 44 or above; Potential risk groups with a total score of 40 to 43; and Normal groups with a score of 39 or below. Our study regrouped high-risk and at-risk group as Problematic Smartphone Use group group and the normal group as the Smartphone Usage Control group. Also, these two groups were performing same tasks during the intervention period. We include this sentence on page 3, in lines 115~119.
Point 6.
Please, explain the incentives that participants received.
Response 6.
Again, thank you for your comments. We explained the incentives that participants received on page 4, in lines 155~161.
Point 7.
The reliability of the scales and questionnaires applied should be provided.
Response 7.
Well confirmed. We have supplemented file for providing questionnaires which were applied in this study and also provide reliability of the scales. Please, refer to page 7, in lines 252~253, 263~264, 273~274 to the changes made.
Point 8.
Statistical analyses need to be improved. A description of the analytic strategy for answering each research question should be provided. Covariates should be considered. To carry out a two-way mixed ANCOVA or moderation and mediation analyses are recommended.
Response 8.
Thank you for providing us such a productive comment. We provided a description of the analytic strategy on page 7~8, in lines 291~297. The difference between LMM and RM ANOVA is that the LMM can be analyzed with covariates. We adopted this method because it can put not only grouping but also various factors such as gender and age into the model. Therefore, in this study, no further covariance analysis was conducted because we added gender as a covariate and observe if it was a significant factor. Also, we presented results on whether there was a time difference between groups except for the effects of gender. Therefore, please consider my response and let me know if we misunderstood your advice.
Point 9.
Some information reported in the result's section should be reported in the method (e.g., lines 237-256; 280-288)
Response 9.
As you suggested, we moved the ‘3.1 Participant Enrollment’ to ‘2.3. Participant Enrollment’ in method section. Thank you for your suggestion, it has improved readability. Please, refer on page 5, in lines 170~193 to the changes made.
Point 10.
Line 249: “Participants who did not complete the assessment were sent notifications to fill up the survey repeatedly and when did not, they were excluded”. Please, provided quantitative data about how many notifications were sent before exclusion. How many participants were excluded for this or other reasons? An intention-to-treat analysis may be conducted.
Response 10.
Another very good comment that you have thankfully provided. During the research period, It was very important to us in encouraging the participants to fill out the survey right time. During the week of 0, 8, 12, from Monday till Sunday, maximum 14 notifications were sent to the participant via SMS until they fill up the survey. We were trying our best to avoid interrupting participants’ daily lives as much as possible. Therefore, we sent the one notification in the morning and sent the other one notification in the afternoon. Despite all this continuing effort, if the participant doesn't respond, they were excluded from the research. We include it on page 5, in lines 183~187.
Among 152 excluded participants:
76 participants were excluded who did not fill up the survey.
50 participants were excluded (those who cannot collect the data less than 75%. It means they nearly turn off the smartphone most of the days).
24 participants were excluded due to the technical Issue.
2 participants were excluded (negatively participated).
Point 11.
There is no discussion, just a summary of the results. A discussion of the implications of the results in terms of substantive findings, and taking into account sources of potential bias and threats to internal and statistical validity, is needed. Moreover, similarities and differences between reported results and previous studies should be included.
Response 11.
We deeply agree with your opinion. According to your advice, I changed the structure of the manuscript (moved from conclusion to the discussion) and included previous researches and potential bias.
Point 12.
Minor issues.
Table 1. Statistics need to be reported. Correct age group data.
Acronym SNS should be defined.
The notation 0.000 needs to be changed by < 0.001
Response 12.
Thank you for your kind information. We reported statistics on page 9, in line 335~338 and correct age group data and defined SNS. Also changed the notation 0.000 by < 0.00

Round 2
Reviewer 2 Report
After revisions the paper could be accepted for publication.
But prior to this:
- The paper could benefit from professional proofreading.
-
The manuscript has a similarity index of over 15 %. Please revise your manuscript to avoid such a high similarity index (e.g., typically, we are okay with an index lower to 15% and no exact sentences from previous publications, excluding quotes). There are papers over 60 words similar.

Author Response
Dear Reviewer,
I really appreciate of your comments and decision that you made. Hereby answering your comments point-by-point.
Point 1. The paper could benefit from professional proofreading.
Response 1. Thank you for your kind advice. According your suggestion, We proofread the manuscript from professional company 'editage'. Please refer the revised manuscript.
Point 2. The manuscript has a similarity index of over 15 %. Please revise your manuscript to avoid such a high similarity index (e.g., typically, we are okay with an index lower to 15% and no exact sentences from previous publications, excluding quotes). There are papers over 60 words similar.
Response 2. Thank you for your consideration on this matter. We revised manuscript and try to avoid overlapped sentences. We lower the similarity of the index over 9 %. Plagiarism was reported due to overlapping headings of the manuscript, so we excluded page numbers, header, and references from the report. Please see the attachment for the Plagiarism check report.
Thank you.

Reviewer 3 Report
I acknowledge the efforts of the authors to improve the manuscript, but there are still some issues unsolved.
Comment 1
Hypotheses need to be stated following the aim of the study.
Comment 2
Please, could you explain the procedure to randomly enrolled the five hundred individuals in the study?
Comment 3
The eligibility of the participants (inclusion and exclusion criteria) should be reported.
Comment 5
Could you explain why the sample sizes of SUC and PSU groups are different? Please, provide the cut-off for S-scale for each group. Mean and standard deviation for each group should also be reported.
Comment 6
Authors incorrectly have induced the reliability of the scores of the measurement instruments. Since reliability is not a stable property, please, report the reliability obtained from the own data and avoid inducing reliability coefficients from other studies.
Comment 7
Please, provide an appropriate reference for the statement in lines 282-284.
Comment 8
Minor issues.
Please, provide statistics in table 1.
Acronym SNS should be defined in line 303.
The notation 0.000 is pending to be changed. Please, change it by < 0.001.
Author Response
Dear Reviewer,
Thank you again for your insightful and valuable feedback. Our manuscripts have been improved through the resolution of your comments one by one. Please find below a point-by-point response to your comments.
Comment 1
Hypotheses need to be stated following the aim of the study.
Response 1
Thank you for your productive comments. We state the hypotheses following the aim of the study. Please refer to page 3, in lines 99~105.
Comment 2
Please, could you explain the procedure to randomly enrolled the five hundred individuals in the study?
Response 2
Thank you for providing us a very good comment. As you suggested, we explained the exclusion procedure of five hundred individuals in the manuscript. Please refer to page 5, in lines 169~177.
Comment 3
The eligibility of the participants (inclusion and exclusion criteria) should be reported.
Response 3
We absolutely agree with your comments. We stated the inclusion and exclusion criteria on page 4, in lines 147~153. Also, as you pointed out in the comment 2, we reported the number of participants who did not eligibility in the study on page 5, in lines 172~177.
Comment 4
Could you explain why the sample sizes of SUC and PSU groups are different? Please, provide the cut-off for S-scale for each group. Mean and standard deviation for each group should also be reported.
Response 4
Another very sharp comment that you have thankfully provided to us. After the screening test for the eligibility of the participant, a total number of 342 participants were conducted a baseline survey (at 0 weeks) through the application (MindsCare). Baseline survey was include the scale which able to measure the problematic smartphone use (the Korean Smartphone Addiction Proneness Scale for Adults; S-scale).
Through the result of S-scale at 0 weeks, we allocated the participants as a SUC and PSU group. Therefore the sample sizes of SUC and PSU groups were different in this study. We provide cutoff for the S-scale on page 7 in lines 249~253. The cutoff for each group are as follows: SUC (total score of 39 or below) and PSU (total score of 40 or above). We reported the Mean and standard deviation for each group on page 7 in lines 251~254.
Comment 5
Authors incorrectly have induced the reliability of the scores of the measurement instruments. Since reliability is not a stable property, please, report the reliability obtained from the own data and avoid inducing reliability coefficients from other studies.
Response 5
Thank you for your comment. We have reported the reliability obtained from our own study data. Apologizes for the confusion. We correct the sentence in the manuscript and also attached the result of reliability analysis, which conducted with our own survey data. Please fine the attached file name ‘reliability analysis.xlsx’.
Comment 6
Please, provide an appropriate reference for the statement in lines 282-284.
Response 6
Well confirmed. We provide reference for the statement in lines 282-284 as follows:
- West, B.T.; Welch K.B.; Galecki A.T. (2014). Linear mixed models: A practical guide using statistical software. CRC Press
- Shek, D. T.; Ma, C. M. Longitudinal data analyses using linear mixed models in SPSS: concepts, procedures and illustrations. ScientificWorldJournal 2011, 5(11), 42-76.
Comment 7
Minor issues.
Please, provide statistics in table 1.
Acronym SNS should be defined in line 303.
The notation 0.000 is pending to be changed. Please, change it by < 0.001.
Response 7
Thank you for your detailed suggestions. We reported statistics on page 9, in table 1. And defined SNS acronym in line 303. The notation of all 0.000 is changed by < 0.001.
